# Determinants of COVID-19 Vaccination Hesitancy among Romanian Pregnant Women

**DOI:** 10.3390/vaccines10020275

**Published:** 2022-02-10

**Authors:** Ioana Mihaela Citu, Cosmin Citu, Florin Gorun, Andrei Motoc, Oana Maria Gorun, Bogdan Burlea, Felix Bratosin, Emanuela Tudorache, Madalin-Marius Margan, Samer Hosin, Daniel Malita

**Affiliations:** 1Department of Internal Medicine I, “Victor Babes” University of Medicine and Pharmacy Timisoara, Eftimie Murgu Square 2, 300041 Timisoara, Romania; citu.ioana@umft.ro; 2Department of Obstetrics and Gynecology, “Victor Babes” University of Medicine and Pharmacy Timisoara, Eftimie Murgu Square 2, 300041 Timisoara, Romania; gorun.florin@umft.ro; 3Department of Anatomy and Embryology, “Victor Babes” University of Medicine and Pharmacy Timisoara, Eftimie Murgu Square 2, 300041 Timisoara, Romania; amotoc@umft.ro; 4Department of Obstetrics and Gynecology, Municipal Emergency Clinical Hospital Timisoara, 1-3 Alexandru Odobescu Street, 300202 Timisoara, Romania; oanabalan@hotmail.com (O.M.G.); bogdanburlea@yahoo.com (B.B.); 5Methodological and Infectious Diseases Research Center, Department of Infectious Diseases, “Victor Babes” University of Medicine and Pharmacy, 300041 Timisoara, Romania; felix.bratosin7@gmail.com; 6Department of Pulmonology, “Victor Babes” University of Medicine and Pharmacy Timisoara, Eftimie Murgu Square 2, 300041 Timisoara, Romania; tudorache_emanuela@yahoo.com; 7Faculty of General Medicine, “Victor Babes” University of Medicine and Pharmacy Timisoara, Eftimie Murgu Square 2, 300041 Timisoara, Romania; marganmm@gmail.com; 8Department of Orthopedics, “Victor Babes” University of Medicine and Pharmacy Timisoara, Eftimie Murgu Square 2, 300041 Timisoara, Romania; samerhosin@gmail.com; 9Department of Radiology, “Victor Babes” University of Medicine and Pharmacy Timisoara, Eftimie Murgu Square 2, 300041 Timisoara, Romania; malita.daniel@umft.ro

**Keywords:** SARS-CoV-2, COVID-19, pregnancy vaccination, vaccination hesitancy, VAX scale

## Abstract

Universal COVID-19 immunization is seen as a critical approach for limiting the spread of SARS-CoV-2 and reducing the danger of new variations emerging in the general population, especially in pregnant women. The literature and accessible research data indicate that vaccination intentions vary greatly by country, with Romania ranking among the European nations with the lowest vaccination rates. Thus, we aimed to investigate the prevalence and extent of COVID-19 vaccine hesitancy among pregnant women in Romania and the factors influencing their decision. A cross-sectional study was conducted on pregnant women referred to the Obstetrics and Gynecology Clinic of the Timisoara Municipal Emergency Hospital in Romania. Participants were asked to complete the validated VAX scale about vaccine hesitancy and to report their willingness to receive a COVID-19 vaccine and their reasons for hesitancy. The group of 184 pregnant women who completed the survey recorded significantly more hesitant respondents than the non-pregnant group with 161 respondents (52.2% vs. 40.3%). They had significantly higher average scores in all VAX scale subsections, while 78.1% of them gave credits to social media for their COVID-19 vaccination decision, compared with 63.0% of non-pregnant women. The independent risk factors for hesitancy were determined as not being afraid of COVID-19 OR = 1.89, below-average income OR = 2.06, trusting social media rumors OR = 2.38, not believing in SARS-CoV-2 existence OR = 2.67, and being a vaccination non-believer OR = 3.15. We advocate for pregnant women to get vaccinated against COVID-19 and for the development of targeted campaigns to address the factors of hesitation. This research emphasizes the critical need for delivering the COVID-19 immunization to the whole community, including pregnant women who may have vaccine-related concerns.

## 1. Introduction

The coronavirus disease pandemic of 2019 (COVID-19) has claimed thousands of lives globally, creating public health issues, overburdening health systems, disrupting supply chains and the economy, and precipitating a mental health crisis [1]. Pregnant and postpartum women may be more prone than non-pregnant women to have a more severe course of COVID-19, and a relationship has been shown between COVID-19 and the risk of preterm and cesarean births [2]. Vertical transmission has also been seen in a few instances in mothers who are SARS-CoV-2-positive, although it is very unusual [3]. Nonetheless, the COVID-19 pandemic substantially impacts expectant women’s well-being, as they worry for their own and their fetus’ health.

Vaccination against infectious diseases is a very effective public health strategy that has been shown to significantly reduce global morbidity and death associated with infection [4]. Numerous vaccines against SARS-CoV-2 infection have been developed and authorized for general population usage, while adhering to all applicable laws. Although pregnant women were not involved in the COVID-19 vaccine development trials, they have had access to these vaccinations since the FDA approved Pfizer/BioNTech immunization for pregnant women in early 2021, followed by the same reaction from the European Medicines Agency [5,6]. Because none of the COVID-19 vaccines contain live viruses or adjuvants that could harm an unborn child, both the American College of Obstetricians and Gynecologists [7] and the Society for Maternal-Fetal Medicine have consistently advocated for the availability of the COVID-19 vaccine to pregnant and lactating women, and both professional societies, as well as the CDC, now recommend vaccination in these populations [8]. In Romania, the decision to promote COVID-19 vaccination during pregnancy was reached later in September 2021, after more data about safety became available.

With an increasing number of pregnant women getting the COVID-19 vaccination throughout their pregnancy and recent studies confirming the safety of the SARS-CoV-2 vaccine in pregnancy [9], disinformation tactics during the COVID-19 pandemic can discourage pregnant women from obtaining it [10]. Therefore, in December 2021, less than 50% of the Romanian population was vaccinated with two vaccine doses [11], and only around 30% of pregnant women currently decide to get vaccinated regardless of obstetricians encouraging them to vaccinate. We believe it is critical to understand the variables that influence vaccine acceptability among different social groups, especially pregnant women, since this will considerably aid in returning the society to its pre-pandemic condition. Therefore, the purpose of this research was to examine pregnant women’s attitudes about COVID-19 vaccination, with a special emphasis on the factors determining COVID-19 vaccine hesitation.

## 2. Materials and Methods

### 2.1. Population Data

A cross-sectional study was performed on pregnant women in the outpatient setting of the Obstetrics and Gynecology Clinic of the Timisoara Municipal Emergency Hospital affiliated with the University of Medicine and Pharmacy from Timisoara, Romania, from 1 October 2021 to 1 December 2021. Patients were informed about the study’s aim and implications, each of them having signed a written informed consent after all their questions had been answered. Our study was conducted in agreement with the Helsinki Declaration Guidelines for scientific experiments involving human subjects, and the Scientific Ethics Committee of the Timisoara Municipal Hospital approved it on 18 October 2021, with approval number No. I-15505/18 October 2021.

### 2.2. Surveys and Variables

A Romanian-translated VAX (Vaccination Attitude Examination) scale was developed to examine anti-vaccination sentiments, which was validated in 2017 [12]. It is a 12-item scale with four subscales. Each question is rated on a scale of 1 to 6, with 1 being a “strongly disagree” answer and 6 representing a “strongly agree” response. Previous research indicates that this survey method has a good level of internal consistency when analyzing vaccination willingness for SARS-CoV-2 [13]. A higher overall score on the VAX scale implies a greater level of anti-vaccination sentiment. The VAX scales can be further classified according to their item numbers: questions 1–3 pertain to mistrust of vaccine benefits; questions 4–6 pertain to concerns about unforeseeable future effects; questions 7–9 pertain to commercial profit concerns; questions 10–12 pertain to a preference for natural immunity.

We used a convenience sampling approach to determine the ideal sample size, which was determined to be at least 221 pregnant women, for a 5% margin of error at a 95% level of confidence and a 30% estimate of the vaccination rate at the time of study. Out of 279 women who agreed to engage in the study and complete our questionnaires, 95 were excluded from the study due to inadequate completion of the survey, leaving 184 pregnant participants. Based on convenience sampling, we calculated the appropriate sample size using a 5% margin of error at a 95% level of confidence for a 30% rate of vaccination in the non-pregnant women at reproductive age, estimating 221 respondents as ideal. The same survey was distributed to 253 non-pregnant women of reproductive age who were in evidence in the outpatient setting of our clinic to determine if pregnancy alone is the main reason for refusing the COVID-19 vaccine, of which, 161 successfully completed it. This study included pregnant and non-pregnant women of reproductive age who were not vaccinated against SARS-CoV-2 throughout the study period, totaling 345 responses. To preserve physical distance and prevent the transmission of COVID-19, participants could examine an online version of the questionnaire that was similar in terms of questions, wording, and presentation sequence. Members of the research team sent questionnaire links to pregnant women, but the online database’s information sorting mechanism was completely automated, and each participant was only allowed to electronically answer questions once. The survey was intended to collect information on the participants’ demographics, including age, place of origin, marital status, level of income, level of education, occupation, and smoking/alcohol consumption behavior. We also surveyed the participants’ trust in the COVID-19 vaccine and conventional vaccines, including a set of additional questions to the VAX questionnaire, such as trusting rumors on social media, having previous unpleasant vaccine side effects, getting insufficient information about vaccines, not being afraid of COVID-19, not believing in the existence of SARS-CoV-2, and lastly, not believing in vaccines. All surveyed questions were categorical with “yes” or “no” answers.

### 2.3. Statistical Analysis

The IBM SPSS software version v26.0 was used to conduct descriptive and inferential statistics. The mean and standard deviation were used to represent continuous data, while absolute and percentage values were used to represent categorical variables. The Student’s *t*-test and the ANOVA tests were used to compare the average values of data analyzed in this study. The VAX scale results were reported as median values and interquartile range (IQR), using the SPSS non-parametric Median test and the pregnancy status as grouping variable. For non-parametric variables, Spearman’s correlation coefficient was calculated, while Pearson’s correlation coefficient was utilized to analyze parametric data. All factors identified in the univariate analysis as having a statistically significant connection with vaccination hesitancy were incorporated in a multivariate backward stepwise logistic regression model. For comparison of proportions between hesitant, unsure, and confident individuals, the Chi-square and Fisher’s tests were employed. The criterion for statistical significance was fixed at alpha = 0.05.

## 3. Results

The surveying period ended with a total of 184 forms completed by pregnant women and 161 completed by non-pregnant women of reproductive age. A comparison of their background revealed similarities in their age, place of origin, marital status, and education level. We found that pregnant respondents had significantly lower income than their non-pregnant peers (63.0% below average income, vs. 50.9%, *p* = 0.023), as there were more numerous unemployed pregnant women (21.1% vs. 11.2%, *p* = 0.012). There were no significant differences in alcohol consumption behavior, although statistically significantly more frequent smokers were in the non-pregnant group (24.2% vs. 10.8%, *p*-value = 0.001). All patients had comparable attitudes towards trusting the SARS-CoV-2 vaccines and other vaccines, with approximately 60% of all respondents not trusting the COVID-19 vaccines, although more than 85% of them trusted other conventional vaccines (Table 1).

The pregnant and non-pregnant study participants shared the VAX questionnaire, along with six other questions regarding reasons likely to determine COVID-19 vaccine refusal. When comparing the total VAX average scores, we observed that pregnant women had significantly higher mistrust levels in the SARS-CoV-2 vaccines (31 vs. 26, *p* < 0.001). Pregnant women scored higher than non-pregnant women in all categories of questions on the VAX survey. More specifically, pregnant women demonstrated higher unfavorable sentiments regarding vaccinations in general than the other group (Questions 1–3). Pregnant women expressed more concern about unforeseen vaccination consequences (Questions 4–6), had more negative views concerning SARS-CoV-2 vaccines (Questions 7–9), and scored higher on health awareness than non-pregnant women (Questions 10–12). Overall, the VAX survey determined that significantly more pregnant women are hesitant than non-pregnant women (52.2% vs. 40.3%, *p* < 0.001). On the other hand, non-pregnant women were more likely to be unsure about accepting the vaccine (35.4% vs. 13.5%). Other reasons for COVID-19 vaccine hesitation are presented in Table 2. Among the six questions addressing other vaccination concerns than those asked in the VAX survey, we observed that trusting rumors on social media had the greatest impact on vaccination hesitancy, with 78.1% of pregnant women answering “yes” to this question, compared with 63.0% of non-pregnant women who responded “yes” (*p* = 0.036). 

The differences among pregnant study participants based on decision factors are described in Table 3. Of the 184 pregnant women, we identified 63 (34.2%) as being confident, 25 (13.5%) unsure, and the majority of 96 (52.1%) as hesitant about COVID-19 vaccines. The trust in social media rumors was the most frequent reason in 125 (67.9%) instances. There was a statistically significant higher trust in social media among hesitant pregnant women (Figure 1) compared with pregnant women (78.1% vs. 53.9%, *p* = 0.013). Another significant contributing factor for SARS-CoV-2 vaccination hesitancy was having experienced previous unpleasant vaccination side effects. Although only 13 (7.1%) pregnant women indicated this reason, significantly more women were unsure (24.0%, *p* = 0.001).

Finally, we attempted to determine the risk factors that contribute to vaccination hesitancy in pregnant and non-pregnant women and identified a rural place of origin, below-average income, trusting rumors on social media, not being afraid of COVID-19, and not believing in the existence of SARS-CoV-2 or vaccines as risk factors (Table 4). Moreover, previous experience of unpleasant vaccine side effects was a risk factor for hesitancy but only in non-pregnant women. Lastly, trusting the SARS-CoV-2 vaccine was determined as a protective factor for vaccination hesitancy in pregnant women (OR = 0.64, CI [0.21–0.86], *p* = 0.042).

After adjusting for risk factors associated with overall hesitancy in the group of pregnant women, we determined that the rural place of origin was an insignificant independent risk factor (AOR = 1.11, CI [1.01–1.35], *p* = 0.062). The statistically significant independent risk factors were, in ascending order by odds, not being afraid of COVID-19, a level of income below average, trusting rumors on social media, not believing in the existence of the SARS-CoV-2 virus, and not believing in vaccines (Table 5).

## 4. Discussion

Our study brings important insights in determining factors for COVID-19 vaccination refusal among pregnant women in Romania. For both groups, sentiments about COVID-19 immunization were more unfavorable than attitudes toward other vaccinations, which is consistent with previous research. In countries such as Israel, acceptance of a COVID-19 vaccine was lower among doctors and nurses than acceptance of seasonal influenza vaccination [14]. U.S.-based research determined that 25% of Americans and 20% of Canadians expressed a willingness to reject a SARS-CoV-2 vaccine, a position consistent with a generally negative attitude toward vaccinations [15], and among nurses in Hong Kong, the primary barrier to accepting the COVID-19 vaccine was the doubt about its safety, efficacy, and effectiveness, while the primary barrier to the flu vaccine was just the doubt about its necessity [16]. We do believe that the sources of information had a powerful impact in determining the hesitancy decision in pregnant women enrolled in this study, since they were more likely to trust the social media sources, as also observed in a large-scale study taking place in Italy in 2018 [17]. The researchers determined that the primary predictors of acceptance of mandated vaccination are information sources and trust in medical experts.

In the current study, we solely investigated vaccination hesitancy among women; thus, the role of gender in vaccine hesitancy could not be determined. Other research found that the female gender was associated with greater vaccine reluctance, which is consistent with the existing research on SARS-CoV-2 vaccines [18]. Thus, the COVID-19 pandemic has emphasized the need to close the gender gap in vaccination reluctance, which has generally been ignored, save for pregnant women. Males were more likely to receive COVID-19 vaccinations, according to an examination of gender roles in vaccine reluctance [19]. For example, a study carried out in the USA on a sample of 672 participants found a 67% overall acceptance of the COVID-19 vaccine. Males, older individuals, Asians, and college and/or graduate degree holders were more likely to accept the vaccine than their counterparts [20]. However, wide demographic and geographical variations in vaccine acceptance for COVID-19 were reported, further highlighting the need for evidence-based community communication to improve the acceptance and effectively respond to the pandemic. The recent conclusion may be appreciated in light of other findings and concerns about COVID-19 quick testing and approval procedure [21]. Specifically, the disparities in views about vaccinations in general and the COVID-19 vaccine, in particular, may be explained in part by COVID-19 vaccine’s quick production, which may have induced vaccine hesitation in both the general population and pregnant women. Despite institutional support for COVID-19 vaccine’s effectiveness and safety, the internet and social media platforms such as Facebook, the most widely used social media platform in Romania, allow anti-vaccination activists to spread misinformation. 

Compared with other populations eligible for COVID-19 vaccination, pregnant women from Romania seemed more influenced by social aspects, such as income, education, and misinformation from social media. Our analysis determined that a lower level of education and a lower income were predictors of vaccination reluctance [22,23], which was similar to prior results from a study conducted in France [24]. This association contrasts with prior research on vaccination that indicated that more educated and wealthy individuals expressed more anxiety about vaccine safety [25,26].

The current research is limited to being a population-based study. Therefore, the results determined by surveying Romanian pregnant and non-pregnant women might not be applicable to women in other regions, whose trust in social media and disbelief in SARS-CoV-2 and COVID-19 vaccines can widely differ. Other country-based features that seem to influence vaccination hesitancy are a rural place of origin, below-average income, and low-level education, which are more prevalent in Romania than in the average countries of the European Union [27]. Other limitations are the online design of the study and failing to meet the appropriate sample size calculated for each group.

## 5. Conclusions

In light of extensive proof of SARS-CoV-2 vaccination safety, the vital need of providing the COVID-19 vaccines extends to the whole population, including pregnant women who may have vaccine-related concerns. Society and social media strongly contribute to vaccination hesitancy among pregnant women in Romania; therefore, we advocate for policy instruments targeting pregnant women to encourage COVID-19 vaccination and for the implementation of a focused campaign to address reluctance reasons.

## Figures and Tables

**Figure 1 vaccines-10-00275-f001:**
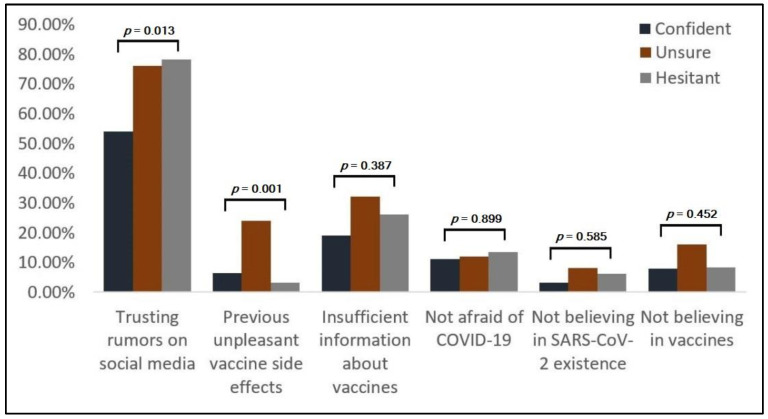
Differences among pregnant study participants based on decision factors.

**Table 1 vaccines-10-00275-t001:** Comparison in baseline characteristics between pregnant and non-pregnant study participants.

Variables *	Pregnant (*n* = 184)	Non-Pregnant (*n* = 161)	*p*
Age	30.6 ± 7.2	29.1 ± 7.8	0.064
**Place of Origin**			0.798
Rural	65 (35.3%)	59 (36.6%)	
Urban	119 (64.7%)	102 (33.4%)	
**Marital Status**			0.256
Married/Concubinage	169 (91.8%)	142 (88.2%)	
Single/Divorced/Widowed	15 (8.2%)	19 (11.8%)	
**Income**			0.023
Below Average	116 (63.0%)	82 (50.9%)	
Above Average	68 (37.0%)	79 (49.1%)	
**Education**			0.794
≤12 years	40 (21.7%)	38 (23.6%)	
>12 years	124 (67.3%)	123 (76.4%)	
**Occupation**			0.012
Employed/Self-Employed	145 (78.9%)	143 (88.8%)	
Unemployed	39 (21.1%)	18 (11.2%)	
**Behavior**			
Frequent alcohol consumption	11 (5.9%)	16 (9.9%)	0.171
Frequent smoker	20 (10.8%)	39 (24.2%)	0.001
**Trusting SARS-CoV-2 vaccine**			0.549
Yes	79 (42.9%)	64 (39.7%)	
No	105 (57.1%)	97 (60.3%)	
**Trusting other vaccines**			0.597
Yes	160 (86.9%)	143 (88.8%)	
No	24 (13.1%)	18 (11.2%)	

* Data reported as *n* (frequency).

**Table 2 vaccines-10-00275-t002:** VAX results and reasons for the hesitancy of the study participants.

Survey Questions *	Pregnant (*n* = 184)	Non-Pregnant (*n* = 161)	*p*
**VAX score, median (IQR)**	31 (8)	26 (9)	<0.001
Questions 1–3 (concerns about trusting vaccines), median (IQR)	7 (3)	5 (3)	<0.001
Questions 4–6 (concerns about unpredictable effects), median (IQR)	10 (5)	9 (4)	0.011
Questions 7–9 (concerns about commercial profits), median (IQR)	6 (3)	4 (2)	<0.001
Questions 10–12 (preference to natural immunity), median (IQR)	8 (4)	7 (3)	0.016
**COVID-19 Vaccination Feeling**			<0.001
Confident	63 (34.3%)	39 (24.2%)	
Unsure	25 (13.5%)	57 (35.4%)	
Hesitant	96 (52.2%)	65 (40.3%)	
**Other reasons for hesitancy**	*n* = 96	*n* = 65	
Trusting rumors on social media	75 (78.1%)	41 (63.0%)	0.036
Previous unpleasant vaccine side effects	3 (3.1%)	2 (3.0%)	0.986
Insufficient information about vaccines	25 (26.0%)	20 (30.7%)	0.511
Not afraid of COVID-19	13 (13.5%)	10 (15.3%)	0.743
Not believing in SARS-CoV-2 existence	6 (6.2%)	4 (6.1%)	0.980
Not believing in vaccines	8 (8.3%)	11 (16.9%)	0.097

* Data reported as *n* (frequency) unless specified differently.

**Table 3 vaccines-10-00275-t003:** Differences among pregnant study participants based on decision factors.

Decision Factors *	Overall (*n* = 184)	Confident (*n* = 63)	Unsure (*n* = 25)	Hesitant (*n* = 96)	*p*
Trusting rumors on social media	125 (67.9%)	34 (53.9%)	19 (76.0%)	72 (78.1%)	0.013
Previous unpleasant vaccine side effects	13 (7.1%)	4 (6.3%)	6 (24.0%)	3 (3.1%)	0.001
Insufficient information about vaccines	45 (24.4%)	12 (19.0%)	8 (32.0%)	25 (26.0%)	0.387
Not afraid of COVID-19	23 (12.5%)	7 (11.1%)	3 (12.0%)	13 (13.5%)	0.899
Not believing in SARS-CoV-2 existence	10 (5.4%)	2 (3.2%)	2 (8.0%)	6 (6.2%)	0.585
Not believing in vaccines	17 (9.2%)	5 (7.9%)	4 (16.0%)	8 (8.3%)	0.452

* Data reported as *n* (frequency).

**Table 4 vaccines-10-00275-t004:** Analysis of risk factors for general vaccination willingness of pregnant women against SARS-CoV-2.

Factors	Pregnant Hesitancy (OR–95% CI)	*p*	Non-Pregnant Hesitancy (OR–95% CI)	*p*
Age	1.00 (0.97–1.04)	0.944	1.00 (0.96–1.03)	0.913
**Place of Origin**		0.011		0.044
Rural	1.67 (1.27–2.38)		1.52 (1.06–1.91)	
Urban	1.18 (0.74–1.89)		1.03 (0.63–1.15)	
**Marital Status**		0.716		0.681
Married/Concubinage ^^^	0.88 (0.36–1.44)		0.94 (0.63–1.48)	
Single/Divorced/Widowed	1.06 (0.82–1.21)		0.97 (0.71–1.09)	
**Income**		0.002		0.039
Below Average	2.52 (1.74–3.08)		2.86 (1.72–3.76)	
Above Average	1.13 (0.93–1.42)		1.05 (0.83–1.46)	
**Education**		0.573		0.418
≤12 years	1.25 (1.04–1.62)		1.30 (1.04–1.78)	
>12 years	0.92 (0.63–1.17)		1.01 (0.85–1.22)	
**Occupation**		0.131		0.294
Employed/Self-Employed ^^^	1.14 (0.86–1.34)		0.90 (0.68–1.20)	
Unemployed	1.36 (0.97–1.59)		1.33 (0.92–1.67)	
**Behavior**				
Alcohol consumption	0.82 (0.39–1.27)	0.728	0.84 (0.32–1.31)	0.661
Smoking	0.75 (0.30–1.14)	0.842	0.78 (0.45–1.24)	0.807
**Reasons for hesitancy**				
Trusting rumors on social media	3.01 (1.84–4.66)	<0.001	2.47 (1.79–3.05)	<0.001
Previous unpleasant vaccine side effects	1.12 (1.01–1.48)	0.057	1.29 (1.07–1.68)	0.040
Insufficient information about vaccines	1.24 (0.90–1.33)	0.146	1.16 (0.84–1.21)	0.243
Not afraid of COVID-19	2.33 (1.29–3.17)	<0.001	2.64 (1.30–3.09)	<0.001
Not believing in SARS-CoV-2 existence	3.43 (2.18–4.51)	<0.001	3.06 (2.11–3.94)	<0.001
Not believing in vaccines	4.05 (2.07–6.42)	<0.001	5.11 (3.27–7.70)	<0.001
**Trusting SARS-CoV-2 vaccine**		0.042		0.066
Yes ^^^	0.64 (0.21–0.86)		0.79 (0.54–1.04)	
No	1.74 (1.13–2.58)		1.66 (1.05–2.83)	
**Trusting other vaccines**		0.094		0.172
Yes ^^^	0.73 (0.49–1.11)		0.92 (0.41–1.43)	
No	1.27 (1.04–1.76)		1.30 (0.91–1.86)	

^^^ Reference category.

**Table 5 vaccines-10-00275-t005:** Adjusted Odds Ratios for factors associated with overall hesitancy in pregnant women.

Factors	Adjusted OR	95% CI	*p*
By rural place of origin	1.11	1.01–1.35	0.062
By below average level of income	2.06	1.74–2.71	0.004
By trusting rumors on social media	2.38	1.90–2.94	<0.001
By not being afraid of COVID-19	1.89	1.54–2.27	0.020
By not believing in SARS-CoV-2 existence	2.67	2.12–3.04	<0.001
By not believing in vaccines	3.15	2.80–3.49	<0.001

OR—Odds Ratio; CI—Confidence Interval.

## Data Availability

The data presented in this study are available on request from the corresponding author.

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
