# Peer review of "Determinants of COVID-19 Vaccination Hesitancy among Romanian Pregnant Women"

_vaccines, 2022, doi:10.3390/vaccines10020275_

Round 1
Reviewer 1 Report
I was invited to revise the paper entitled "Determinants of COVID-19 Vaccination Hesitancy Among Romanian Pregnant Women". It was a cross sectional study aimed to investigate factors associated to COVID19 vaccine uptake among Romanian pregnant women.
The topic is interesting but it needs several improvements:
- Sample size estimation method should be better specified (alpha error, expected acceptance rate and power);
- the validation of the questionnaire should be reported;
- It is unclear as non-pregnant women were selected. It can represent a bias;
- non parametric tests as Mann-Whitney and Kruskall Wallis, do not compare median!!! They compare ranks!!! It is a common error!!!
- Scores are discrete variables and should be reported as median and IQR;
- In discussion section, Authors should compare their results with one of the most important study on Vaccine Hesitancy among pregnant women performed in EU (NAVIDAD study from Italy - 10.1016/j.vaccine.2018.04.029 )
- Backward selection in multivariate logistic selection was set ad 0.05? Usually it is set at 0.10;
- As supplementary analysis I sugget to analyze factors associated to Overall Hesitancy, using pregnancy status as covariate;
Author Response
I was invited to revise the paper entitled "Determinants of COVID-19 Vaccination Hesitancy Among Romanian Pregnant Women". It was a cross sectional study aimed to investigate factors associated to COVID19 vaccine uptake among Romanian pregnant women.
The topic is interesting but it needs several improvements:
- Sample size estimation method should be better specified (alpha error, expected acceptance rate and power);
- the validation of the questionnaire should be reported;
- It is unclear as non-pregnant women were selected. It can represent a bias;
- non parametric tests as Mann-Whitney and Kruskall Wallis, do not compare median!!! They compare ranks!!! It is a common error!!!
- Scores are discrete variables and should be reported as median and IQR;
- In discussion section, Authors should compare their results with one of the most important study on Vaccine Hesitancy among pregnant women performed in EU (NAVIDAD study from Italy - 10.1016/j.vaccine.2018.04.029 )
- Backward selection in multivariate logistic selection was set ad 0.05? Usually it is set at 0.10;
- As supplementary analysis I sugget to analyze factors associated to Overall Hesitancy, using pregnancy status as covariate;
Dear reviewer,
We all appreciate your feedback and the time taken to evaluate our manuscript. In order to improve our paper, we made the following edits based on your advice, in addition to the other reviewers:
Line 107: added the margin of error based on which the sample size was calculated. We couldn’t estimate the expected acceptance rate.
Lines 96 and 100: we specified the validation of the questionnaire in use
The non-pregnant patients were considered as the control group to observe if their behavior is consistent with pregnancy status. If you were meaning how the non-pregnant women were selected, we added a short line at 111, and we also describe between lines 118 and 121 how the surveying technique took place.
Thank you for letting us know about this common error in reporting median differences with Mann-Whitney and Kruskall-Wallis tests. However, these tests were mentioned in the materials and methods, but in fact we did not have any median values or ranks to compare. Therefore, we removed this part at lines 133-135.
Indeed, we were also thinking to report the scores of the VAX scale as median and IQR, but we saw that it was validated with data being reported as average value and standard deviation (please see the newly added references 12 and 13.
We included the recommended study in the discussions section, as we believe it matches our results of reluctancy towards vaccination in pregnant women that manifest trust issues due to anti-vaccination campaigns rolling on the social media (Lines 217-222).
Yes, the selection in multivariate regression was set by default at 0.05.
Thank you for your suggestion, we will consider.
Best regards

Reviewer 2 Report
This study aims to determine whether vaccination hesitancy is higher in pregnant women than in non-pregnant women in Romania.
In line 52, the word “moms” is used. This is an informal word and should not appear in an academic article.
The introduction refers to the support for vaccination against COVID 19 in pregnant women by American scientific societies, the FDA, and the CDC. Still, surprisingly no reference is made to European health authorities. A reference to the position of the EMA should be included.
https://www.ema.europa.eu/en/news/covid-19-latest-safety-data-provide-reassurance-about-use-mrna-vaccines-during-pregnancy
In addition, the reference we provide to the European Medicines Agency includes additional literature on vaccination against COVID 19 in pregnant women.
The material and methods indicate that a Romanian translation of the validated VAX scale was used. No bibliographic reference is provided. The authors should tell whether it is a scale they have developed themselves or a translation of an existing scale.
If it is a translation of an existing scale into Romanian, the bibliographic reference of the original scales should be provided. The translation system should be explained.
If the authors developed a new scale, the questionnaire should be presented in an annex to the article. Explaining how the validation has been performed, the dimensions of the scale, and its interpretation.
It should be better explained the sample size how the figure of 221 women, with 95% confidence, is arrived. Usually, when calculating the sample size, the confidence and the estimate of the parameter in the population and the maximum error you are willing to admit should be taken into account. It is explained that the sampling procedure for pregnant women is convenience sampling. It is not indicated in the selection procedure of the controls. It should be specific in the article.
English has to be improved. For example, line 113 incorrectly uses the auxiliary verb may when the past tense of may is might.
Table two should be modified so that it is possible to know what is defined in the questions. A reader who does not have access to the questionnaire cannot interpret it. Reference should be made to the content of the questions, not to the numbering of the questions in the questionnaire.
The text and tables are inconsistent. Sometimes “p” is used, and sometimes “p-value” is used.
It would be best if you always used “p”.
Author Response
This study aims to determine whether vaccination hesitancy is higher in pregnant women than in non-pregnant women in Romania.
In line 52, the word “moms” is used. This is an informal word and should not appear in an academic article.
The introduction refers to the support for vaccination against COVID 19 in pregnant women by American scientific societies, the FDA, and the CDC. Still, surprisingly no reference is made to European health authorities. A reference to the position of the EMA should be included.
https://www.ema.europa.eu/en/news/covid-19-latest-safety-data-provide-reassurance-about-use-mrna-vaccines-during-pregnancy
In addition, the reference we provide to the European Medicines Agency includes additional literature on vaccination against COVID 19 in pregnant women.
The material and methods indicate that a Romanian translation of the validated VAX scale was used. No bibliographic reference is provided. The authors should tell whether it is a scale they have developed themselves or a translation of an existing scale.
If it is a translation of an existing scale into Romanian, the bibliographic reference of the original scales should be provided. The translation system should be explained.
If the authors developed a new scale, the questionnaire should be presented in an annex to the article. Explaining how the validation has been performed, the dimensions of the scale, and its interpretation.
It should be better explained the sample size how the figure of 221 women, with 95% confidence, is arrived. Usually, when calculating the sample size, the confidence and the estimate of the parameter in the population and the maximum error you are willing to admit should be taken into account. It is explained that the sampling procedure for pregnant women is convenience sampling. It is not indicated in the selection procedure of the controls. It should be specific in the article.
English has to be improved. For example, line 113 incorrectly uses the auxiliary verb may when the past tense of may is might.
Table two should be modified so that it is possible to know what is defined in the questions. A reader who does not have access to the questionnaire cannot interpret it. Reference should be made to the content of the questions, not to the numbering of the questions in the questionnaire.
The text and tables are inconsistent. Sometimes “p” is used, and sometimes “p-value” is used.
It would be best if you always used “p”.
Dear reviewer,
We all appreciate your feedback and the time taken to evaluate our manuscript. In order to improve our paper, we made the following edits based on your advice, in addition to the other reviewers:
Line 52: we changed the word to “mothers”
Line 62: we added the EMA reference regarding the decision
Line 96 and Line 100: The VAX scale was translated in Romanian, and we referenced its validation along with a study that validated the internal consistency of the same scale in the setting of COVID-19 vaccines (references 12 and 13).
Line 124-128: we did not develop a new scale, although six additional questions complemented the scale to assess an opinion. They did not use a likert scale type, but just a yes or no. If necessary, we will attach the additional questions as a separate annex.
Line 107-108 and 111: we specified the margin of error (5%) for which the sample size was calculated, considering a proportion of approximately 30% of pregnant women willing to vaccinate. We reached this number based on the vaccination rate in Romania at the time of study.
Line 117 (previously line 113): We used a software to check grammar and spelling mistakes since we are not native English speaker. Although if you consider necessary, we will submit our manuscript to a specialized English editing service.
We edited Table 2 by including the meaning of each question block.
We changed p-values to “p”, as recommended.
Best regards

Round 2
Reviewer 1 Report
I was invited to review the revised version of the paper entitled "Determinants of COVID-19 Vaccination Hesitancy Among Romanian Pregnant Women". Despite Authors responded propely to some observations, there are some point that were not addressed:
- Authors have to specify the selection criteria and the selection methods of non-pregnant women. It can represent a selection bias and should be deeply described;
- Scores have to be reported as median and IQR;
- The required supplementary analysis was not performed.
Author Response
I was invited to review the revised version of the paper entitled "Determinants of COVID-19 Vaccination Hesitancy Among Romanian Pregnant Women". Despite Authors responded propely to some observations, there are some point that were not addressed:
- Authors have to specify the selection criteria and the selection methods of non-pregnant women. It can represent a selection bias and should be deeply described;
- Scores have to be reported as median and IQR;
- The required supplementary analysis was not performed.
Dear reviewer,
Thank you once again for taking the time to give us valuable feedback. Please consider the following answers:
- We have described the selection criteria for non-pregnant women, which were the same methods used for the pregnant group, although the proportions vary. By definition itself, the convenience sample method is flawed compared with a probability sampling, since it is more likely to give biased results and not representing correctly the entire population addressed by the survey. However, we considered the convenience method sufficient for the purpose of this study. Overall, we believe the Materials and Methods section covers all important steps in selection criteria for both groups, including the margin of error, confidence interval, study populations, estimates for vaccination, and the patients belonging to our outpatient clinic.
- We have calculated the VAX scale results that were reported as median values and interquartile range (IQR), using the SPSS non-parametric Median test and the pregnancy status as grouping variable (please see Table 2 and Lines 136-138).
- We appreciate your recommendation to analyze factors associated to Overall Hesitancy, using pregnancy status as covariate, but due to time constraints and busy schedules we are unable to perform another statistical analysis, add a new section in the body of the manuscript and basically edit all the other section to include the new analysis.
Best regards

Round 3
Reviewer 1 Report
I was invited to review the revised version of the paper entitled "Determinants of COVID-19 Vaccination Hesitancy Among Romanian Pregnant Women".
Authors declared to have no time to add requested analyses. These are words that an Editor and a Referee does not wanto to see.
The Editor will decide about this point
Author Response
Dear reviewer,
Thank you for your message. We are sorry for the misunderstanding created.
After carefully discussing with the authors all comments addressed to us, we understood the comment about risk factor analysis. Although initially we were somewhat confused about the additional analysis, we realized it was all about an independent risk factor analysis only for overall hesitancy only in the group of pregnant women. Therefore, we calculated the adjusted odds ratios for all factors that were determined to be statistically significant in the unadjusted regression model.
The additional changes are as follow:
- Lines 39-43: The abstract section was modified according to the additional analysis
- Several small corrections were made at lines 81, 84, 201-205.
- Lines 209-217: Table 5 and interpretation were added as suggested.
Please consider our sincere apology,
The authors

Round 4
Reviewer 1 Report
The paper is know acceptable for publication.
minor comment: line 210 Only CI was reported. Please add the OR.
Author Response
Dear reviewer,
Thank you very much for your favourable feedback.
We corrected the manuscript accordingly, and we appreciate this collaboration that helped our manuscript meet the quality standards.
Best regards
